# Haste Makes Waste: Teaching Image Restoration to Learn Distributions from Pixels to Patterns

## Abstract

In this paper, we revisit the image restoration (IR) task and propose a new training strategy that models the IR problem as a distribution mapping challenge from two perspectives, i.e., (1) the intra-pixel regression and (2) the inter-pixel interaction. At the beginning of optimization, due to the pattern distribution involving a group of pixels within a neighborhood, it is not very easy for the model to capture such multi-pixel distribution mapping. A more optimal solution would be firstly teaching the model to learn a relatively simple yet important distribution w.r.t the pixel-by-pixel mapping between the degraded/clean pixels, as warming up. By doing so, the learned distribution is served as a prior, regarded as an injection of a kind of inductive bias into the model's whole optimization procedure. Subsequently, as conventional, the model is shifted to focus on the mapping distribution of the cross-pixel patterns, which ensures the consistency and fidelity of the image patterns. The final learned mapping is a joint distribution, which transfers the knowledge from the pixel distributions to the pattern ones. Experimental results indicate that under the compact and elegant training paradigm, the newly learned joint distribution is closer to the ideal one and yields a stronger representation ability, to circumvent the dilemma of the difficulty for existing methods to learn the patterns mapping distribution between degraded/clean images right off the bat.

## 1 Introduction

Image restoration (IR), aims to remove degradations (e.g., blur and noise) from a low-quality degraded image (LQ) to generate a high-quality counterpart (HQ). Recent years have witnessed great success in IR tasks with deep learning techniques (Nah et al., 2017; Vincent et al., 2008). Furthermore, many existing methods face challenges either in designing complex network architectures (Liang et al., 2021; Li et al., 2023b; Chen et al., 2023b) or in incorporating auxiliary structures like (Liu et al., 2022; Yu et al., 2018; Chen et al., 2024). These approaches aim to enhance the models' representational power and generalization ability but often introduce increased complexity and external dependencies. Some approaches (Wang et al., 2022; Zamir et al., 2022a; Li et al., 2021) propose new training strategies also to improve the average performance on benchmark datasets.

The essence of the IR problem is to learn the pixel-to-pixel regression mapping from the degraded image space to the clean one. However, as illustrated in Fig. 1, rather than learning an easier pixel-by-pixel distribution mapping, current IR methods generally apply an across-the-board training strategy to their optimization process that makes the model directly learn a distribution mapping involving a set of pixels within a neighborhood from stem to stern. They consistently overlook the fact that networks struggle to effectively model the cross-pixel contents and patterns distribution mapping, which need to take multi-pixels into account at a time in the early training stage. Therefore, this increases the overall complexity of training the IR model and further results in the optimization being trapped in a sub-optimal geometry landscape.

In this paper, we identify the nature of pixel-to-pixel regression for IR tasks and model it as an optimization problem from two perspectives: (1) the intra-pixel regression and (2) the inter-pixel interaction. With such observations, we propose the In**TRA**-patch **P**ixel-**S**huffle training strategy, dubbed **TRAPS**. Specifically, from the intra-pixel perspective, at the early stage in training, the

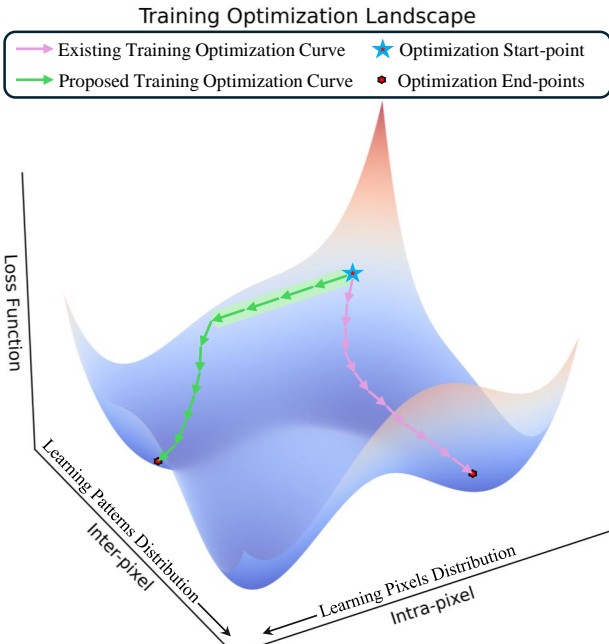

Figure 1: We model the image restoration task as an optimization problem from two perspectives, i.e., first learn the pixel distribution mapping and then learn the pattern distribution mapping. When optimizing the IR problem in the early stage, it is unrealistic to expect the network to right off the bat capture the mapping of content/pattern distributions within pixel neighborhoods between the degraded-clean image space, which would be counterproductive to the overall optimization procedure. While the IR task is essentially a pixel-by-pixel regression problem, if the network can be guided to prioritize learning a simpler representation, i.e., a single-pixel-to-single-pixel mapping between degraded-clean pixel space (as shown by the light-green band in the figure, the optimization is only along the 'intra-pixel' axis at this situation), the whole training process would be able to escape from the original geometry landscape and converge to another better one.

model's premature focus on the cross-pixel pattern distribution between LQ and ground truth (GT) is half-hearted for the IR problem. This is because, in the warm up stage of training, the model has a limited ability to directly capture such pattern distribution (Ye et al., 2022). However, instead of involving capturing the distribution among multi-pixels at a time, it would be a much better solution to allow the neural network to first learn the pixel-by-pixel distribution mapping at this stage since this process is to discover a single-pixel-to-single-pixel mapping relation without requiring attention to semantic information and local content patterns. To achieve this, we randomly shuffle the pixels of an image batch according to the pre-generated indices (similarly on GTs) and feed them to the model to learn this pixel-by-pixel distribution preferentially. This operation serves as a training warm up strategy and plays a crucial role in our overall training paradigm. The proper warm up preparation helps avoid a prolonged and exhausting training overhead. Subsequently, we early-stop to learn the prior distribution and transition the model's training to learn the joint one, which allows the model begin to learning pattern distribution mapping across pixels. Compared to the regular training strategy, this concise modification can better optimize the primitive restoration problem and boost the model performance.

The contributions of our work can be summarized as:

- We re-visit the IR task, identifying its pixel-to-pixel regression essence, and model it as an optimization problem from the intra-pixel and inter-pixel perspectives.

- We propose a new training strategy for the IR task. This strategy is tailored from the two perspectives observations and can be regarded as a free data augmentation strategy or a training warm up approach.

- Our proposed strategy can be seamlessly incorporated into existing supervised IR methods in a zero-burden manner. It essentially introduces a kind of inductive bias into the models, which can easily boost their performance.

## 2 RELATED WORK

### 2.1 IMAGE RESTORATION

Image restoration aims to recover images degraded by factors like noise and blur. Driven largely by the capabilities of various backbone neural networks architecture (Krizhevsky et al., 2012; Dosovitskiy et al., 2021) with the development of deep learning, significant advancements have been made in sub-fields such as image denoising (Zhang et al., 2017; 2019; Zamir et al., 2022b) and image deblurring (Nah et al., 2017; Xu et al., 2014). Generally, IR methods like (Liang et al., 2021; Zamir et al., 2022a; Wang et al., 2022; Li et al., 2023b) use a data-driven approach after the network structure is designed and train directly on pairs of data until the network converges. These methods do not pay attention to the fact that IR tasks should have different preferences when dealing with inputs with different content distributions. However, even if some methods (Gu et al., 2024; Jiang et al., 2023a) pay attention to this issue, they still do not recognize the inherent property of IR task pixel-to-pixel regression. This entangles the intra- and inter-processes and leads to sub-optimal training strategies.

### 2.2 TRAINING STRATEGY ON IMAGE RESTORATION

Recently, researchers have explored various deep learning-based training strategies for IR tasks. RL-Restore (Yu et al., 2018) introduces reinforcement learning into IR. Huang *et al.* presents a complex training strategy, BayerEnsemble (Huang et al., 2022), which addresses raw image denoising based on the Bayer filter. Liu *et al.* (Liu et al., 2022) and Chen *et al.* (Chen et al., 2024) leverage pre-training strategies to learn generalized natural image priors for IR tasks. DIL (Li et al., 2023a) proposes a causality-based training strategy to improve the generalization ability of IR methods for unknown degradations. Chen *et al.* introduce MaskedDenoising (Chen et al., 2023a) to explicitly learn representations for content reconstruction in image denoising from a masked image modeling perspective. Jiang *et al.* introduce a few-shot learning paradigm (Jiang et al., 2023b) to eliminate the reliance on clean-noisy image pairs for the denoising task. Self-supervised training is introduced to DCD-Net (Zou et al., 2023) to progressively achieve improved denoising results by an iterative optimization strategy. Huang *et al.* propose WaveDM (Huang et al., 2024) to learn the low-/high-frequency spectrums in distinct modules in the transform domain for IR tasks by introducing the Diffusion Model (Ho et al., 2020). The performance gains from such training strategies come at the cost of introducing complex computational mechanisms or additional auxiliary structures that bring an additional burden into the training process.

## 3 METHOD

### 3.1 MOTIVATION: LEARNING DISTRIBUTIONS FROM PIXELS TO PATTERNS

Recently, there has been a proliferation of IR methods based on deep learning, which have achieved considerable performance on various benchmark datasets. However, these previous approaches have consistently ignored one of the fundamental properties of the IR task, i.e., pixel-by-pixel regression, instead choosing to optimize it from a more complicated perspective - learning the cross-pixel pattern distribution. Therefore, we re-visit the IR task from this perspective and raise the following scientific question: on the basis of the existing network structure and the available data, is it possible to take advantage of the pixel-by-pixel regression and re-model the IR optimization problem to find a concise and elegant training strategy to achieve better representation ability?

To this end, we seek the probabilistic distribution modeling for help. The essence of the IR problem is to learn a mapping whereby the model is expected to map images from one distribution (degraded image space) into another image space (clean image space). Formally, hypothesizing that the ideal clean image $\mathcal{Y}_{ideal}$ has the distribution $q_{ideal}$, i.e., $\mathcal{Y}_{ideal} \sim q_{ideal}$. The original IR methods sample training data points from the degraded image space $\mathcal{X}_{in} \sim q_{degraded}$ and try to directly learn the mapping relation from $\mathcal{X}_{in}$ to the clean image distribution $q_{ideal}$. This approach removes degradation by mapping limited degradation from $\mathcal{X}_{in}$ to the clean image space partially. Due to the large gap between the two distributions, however, direct modeling $q_{degraded}$ to $q_{ideal}$ is more difficult, and

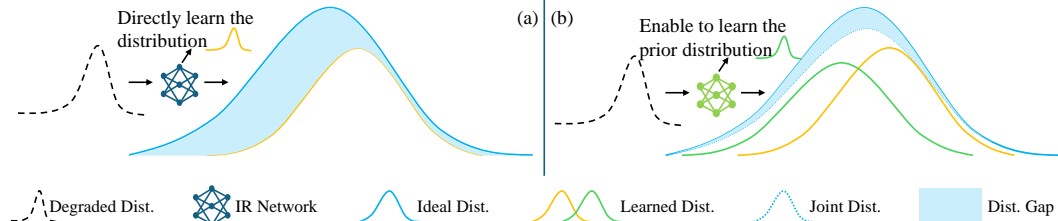

Figure 2: Clarifying our motivation from the distribution perspective. The essence of IR problems is to model the mapping of distributions from low-quality image space to a clean one. (a): Existing methods attempt to learn the mapping relationship between these two distributions directly through ground-truth supervised signaling. Due to the large gap between the two distributions, this approach only leads the network to learn a compromise mapping distribution. (b): Provided that the available data samples and the network structure remain static, to bridge the discrepancy between the two distributions, we try to guide the network to learn a priori distribution first. The new distribution the model learned can be observed as a joint distribution, which smoothly transfers the distribution from the pixel level to the pattern one. This will make the final distribution closer to the ideal one.

these methods often manage to learn a compromise distribution, which can be denoted as Eq. 1:

$$q(\mathcal{X}_{out}|\mathcal{X}_{in}) \tag{1}$$

The learning process for this distribution mapping is shown in Fig. 2(a). To reduce the complexity of learning the entire distribution mapping, we first guide the model to learn a priori distribution $q(\mathcal{Y}_{p2p}|\mathcal{X}_{p2p})$, which enables the model to initially focus on pixel-to-pixel distribution mapping in IR problems. Once the model has learned this prior distribution, we then transition to allowing it to learn the mapping from the degraded image distribution to the clean image distribution, as shown in Fig. 2(b). The distribution mapping of this process can be denoted as Eq. 2:

$$q(\tilde{\mathcal{X}}_{out}|\mathcal{X}_{p2p}, \mathcal{X}_{in}) \tag{2}$$

The final learned distribution can be considered as a joint one from the two distributions. Benefiting from the prior distribution first learned, the joint distribution remains a smaller gap to the ideal distribution $q_{ideal}$, i.e., we have:

$$||q_{ideal} - q(\tilde{\mathcal{X}}_{out}|\mathcal{X}_{p2p}, \mathcal{X}_{in})|| < ||q_{ideal} - q(\mathcal{X}_{out}|\mathcal{X}_{in})|| \tag{3}$$

### 3.2 TECHNICAL DETAILS: THE INTRA-PATCH PIXEL-SHUFFLE STRATEGY

In this section, we detail the specifics of our proposed solution and answer the research question posed in the motivation above. In response to that, we propose the In**TRA**-patch **P**ixel-**S**huffle training strategy, dubbed **TRAPS**. The comprehensive framework is elucidated in Fig. 3.

Specifically, considering the high difficulty of directly learning the distribution $q(\mathcal{X}_{out}|\mathcal{X}_{in})$ - the model's limited ability to capture cross-pixel content and pattern representations at the early stages of training, we prioritize a simpler yet crucial mapping relationship. That is, we first enable the model to learn a priori distribution $q(\mathcal{Y}_{p2p}|\mathcal{X}_{p2p})$ on how to map individual pixels from the LQ image distribution into the pixel space of the corresponding GT image distribution. To implement this, before each iteration in the early stages of training, we randomly shuffle the pixels of each image batch - taking the input LQ as an example, we first reshape the LQ from $B, C, H, W$ to $B, C, H * W$ along its pixel dimensionality and further generate a corresponding randomly-indexed tensor with the same shape, $shuffle\_indices$, where each slice of $shuffle\_indices$ is populated by a random permutation consisting of 0 to $HW-1$, for a total of $B*C$ such slices. Once the $shuffle\_indices$ are obtained, we then shuffle the pixels of each image in the LQ according to the random indices in the tensor to obtain $Shuffled\_LQ$. Similarly, by doing the above for GT we obtain $Shuffled\_GT$. The gathered $Shuffled\_LQ$ will be subsequently fed to the IR network for iterative optimization under the supervision of the signal $Shuffled\_GT$ to allow the model to capture the priori distribution $q(\mathcal{Y}_{p2p}|\mathcal{X}_{p2p})$ until it is asked to stop early. At this point, we consider the network's warm up period complete, i.e., the priori distribution has been learned, having developed the ability to learn cross-pixel content and pattern distributions. The network then naturally transitions to the

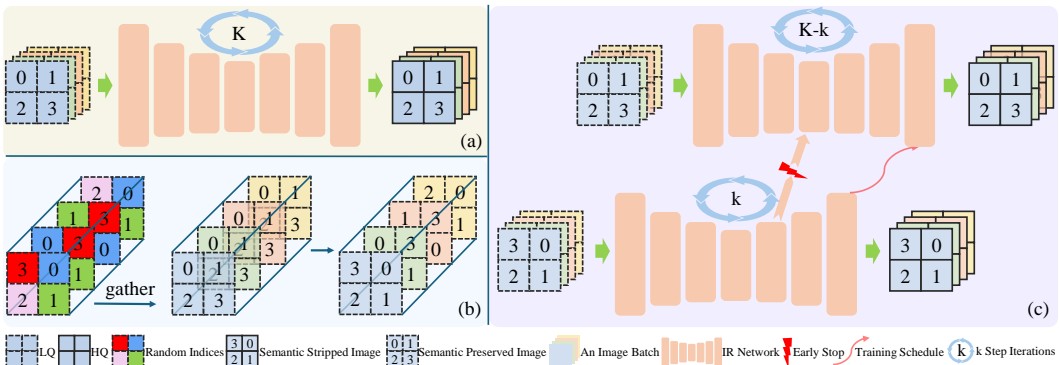

Figure 3: The overview of our proposed training strategy. (a) denotes the conventional training paradigm of current existing IR methods, which optimize the IR problem by engaging the model to learn the mapping between pattern distributions in the neighborhoods of input and output pixels directly. (c) represents our proposed training paradigm. We re-model the IR task as an optimization problem from the two perspectives: intra-pixel regression and inter-pixel interaction. It is not so easy to make the network learn the cross-pixel distribution mapping early in the training. Therefore, we first make the model try to learn a simpler mapping between individual pixels, so as to get a more robust training warm up preparation for the whole optimization process. After that, the optimization schedule can smoothly transition from learning the mapping representations between individual pixels to learning the mapping representations of the pattern distributions across pixels. (b) An intuitive demonstration of how to get semantic stripped images (after the pixels are randomly shuffled) from semantic preserved images (the original images).

next phase, where it continues to learn the distribution $q(\tilde{\mathcal{X}}_{out}|\mathcal{X}_{p2p}, \mathcal{X}_{in})$ refine this representation ability until the end of the training schedule.

A more formalized formulation is as follows: consider the current training task has a total of $K$ iterations, the image restoration model is denoted by IRNet, and the expected output of the network after receiving the input LQ is HQ. The original training strategy and ours can be denoted by Eq. 4,

$$\text{HQ}_i = \text{IRNet}(\text{LQ}_i), i \in \{0, 1, \cdots, K{-}1\} \tag{4}$$

and Eq. 5, respectively,

$$\begin{cases} \text{Shuffled\_HQ}_i = \text{IRNet}(\text{Shuffled\_LQ}_i) & , i \in \{0, 1, \cdots, k{-}1\}, \\ \text{HQ}_i = \text{IRNet}(\text{LQ}_i) & , i \in \{k, k{+}1, \cdots, K{-}1\}. \end{cases} \tag{5}$$

where the subscript $i$ denotes the $i$-th step of the total iterations $K$, and $k$ means the iteration step when the intra-pixel training phase early-stops.

## 4 EXPERIMENTS AND RESULTS

The main experimental results are first reported in this section. We conduct extensive experiments on two representative IR tasks, i.e., image denoising and image deblurring, to verify the effectiveness of the proposed training strategy. Due to the zero-burden integration property of our method, for a fair comparison, we retrain all the models with the same hyperparameter configurations and training protocols and further report the quantitative metrics and visual qualitative results on widely available benchmark test sets.

### 4.1 IMAGE DENOISING

We first incorporate the proposed training strategy into the most current state-of-the-art methods: SwinIR (Liang et al., 2021), HAT (Chen et al., 2023b) and GRL (Li et al., 2023b) and investigate the performance of the commonly used testsets, Kodak24 (Franzen, 1999), Mcmaster (Zhang et al., 2011), CBSD68 (Martin et al., 2001) and Urban100 (Huang et al., 2015) on the Gaussian image denoising. The metrics: PSNR, SSIM (Wang et al., 2004), and LPIPS (Zhang et al., 2018), are reported.

Figure 4: Visual comparison of **color** and **grayscale** Gaussian image denoising on Urban100 (Huang et al., 2015) and Kodak24 (Franzen, 1999) with noise level $\sigma = 50$. Zoom in for more details.

As shown in Tab. 1, compared with the previous state-of-the-art methods, our training strategy can consistently improve the performance w.r.t. the metrics across all the datasets. Noting that, although our strategy does not introduce any additional complicated mechanisms and auxiliary architectures, it can even improve some methods by more than 0.1dB on more difficult datasets at zero-burden cost, e.g., improving GRL-S by 0.14dB on Urban100.

Table 1: Quantitative results about color image denoising. The best results are highlighted. The PSNR and SSIM are reported on the RGB and Y channel, respectively. The AWGN noises with level $\sigma = 50$ are added to clean images.

| Method | McMaster | | | Urban100 | | | CBSD68 | | | Kodak24 | | |
|---|---|---|---|---|---|---|---|---|---|---|---|---|
| | PSNR(↑) | SSIM(↑) | LPIPS(↓) | PSNR(↑) | SSIM(↑) | LPIPS(↓) | PSNR(↑) | SSIM(↑) | LPIPS(↓) | PSNR(↑) | SSIM(↑) | LPIPS(↓) |
| SwinIR-B | 28.88 | 0.8645 | 0.2794 | 27.57 | 0.8690 | 0.2421 | 27.79 | 0.8144 | 0.3047 | 28.77 | 0.8219 | 0.3428 |
| **+ TRAPS** | 28.94 | 0.8662 | 0.2749 | 27.65 | 0.8707 | 0.2402 | 27.83 | 0.8167 | 0.3008 | 28.81 | 0.8253 | 0.3360 |
| SwinIR-S | 28.17 | 0.8526 | 0.3066 | 26.88 | 0.8523 | 0.2705 | 27.45 | 0.8033 | 0.3217 | 28.30 | 0.8106 | 0.3627 |
| **+ TRAPS** | 28.25 | 0.8542 | 0.3045 | 26.95 | 0.8543 | 0.2684 | 27.49 | 0.8041 | 0.3198 | 28.36 | 0.8118 | 0.3601 |
| GRL-S | 28.89 | 0.8623 | 0.2853 | 27.52 | 0.8650 | 0.2495 | 27.74 | 0.8105 | 0.3082 | 28.71 | 0.8181 | 0.3487 |
| **+ TRAPS** | 28.99 | 0.8640 | 0.2812 | 27.66 | 0.8678 | 0.2447 | 27.80 | 0.8117 | 0.3065 | 28.79 | 0.8202 | 0.3458 |
| HAT-S | 28.84 | 0.8620 | 0.2839 | 27.41 | 0.8640 | 0.2510 | 27.72 | 0.8108 | 0.3065 | 28.66 | 0.8184 | 0.3480 |
| **+ TRAPS** | 28.90 | 0.8646 | 0.2767 | 27.52 | 0.8678 | 0.2444 | 27.78 | 0.8148 | 0.2988 | 28.71 | 0.8220 | 0.3387 |

We further show the visual results for image denoising in Fig. 4. It can be observed that with the incorporation of our strategy, these methods are able to remove more noise while preserving sharper texture details and enhancing perceptual quality.

## 4.2 IMAGE DEBLURRING

Besides the experiments on image denoising, we also investigate the effectiveness of our training strategy on image deblurring. We seamlessly integrate our method into the current state-of-the-art sota methods: GRL (Li et al., 2023b), HAT (Chen et al., 2023b), MIRNet (Zamir et al., 2022b) and SwinIR (Liang et al., 2021). And we validate the effectiveness of our method on the widely used image deblurring dataset: GoPro (Nah et al., 2017), HIDE (Shen et al., 2019) and real-world blurred dataset RealBlur (Rim et al., 2020). Note that we only train the models on GoPro (Nah et al., 2017) dataset and validate **directly** on other test sets, i.e., we do not train a specific model for the data distribution of RealBlur (Rim et al., 2020). The PSNR and SSIM are reported.

Tab. 2 shows the experimental results for single image motion deblurring on synthetic datasets Go-Pro (Nah et al., 2017), HIDE (Shen et al., 2019) and real-world blurred dataset RealBlur (Rim et al., 2020), respectively. It can be observed that our training strategy can consistently provide higher PSNR and SSIM values for the original method. Note that when the original method has a smaller number of parameters, i.e., the method itself has a weaker fitting ability, our strategy can bring more significant performance improvement to it. For example, it can improve MIRNet (Zamir et al., 2022b) by 0.xx dB on HIDE (Shen et al., 2019)). This observation also demonstrates that our approach can be regarded as a data-augmentation strategy.

Visual results are shown in Fig. 5. We can see that the image produced by our method is sharper and visually closer to the ground truth than those of the original algorithms.

Figure 5: Visual comparisons of the **single-image motion deblurring** on GoPro (Nah et al., 2017) (1st row), HIDE (Shen et al., 2019) (2nd row), and RealBlur (Rim et al., 2020) dataset (3rd row) . Zoom in for more details.

Table 2: Quantitative results about single image motion deblurring. The best results are highlighted. The PSNR and SSIM are reported on the RGB channel and Y channel, respectively.

| Method | GoPro | | HIDE | | RealBlur-R | | RealBlur-J | |
|---|---|---|---|---|---|---|---|---|
| | PSNR(↑) | SSIM(↑) | PSNR(↑) | SSIM(↑) | PSNR(↑) | SSIM(↑) | PSNR(↑) | SSIM(↑) |
| SwinIR-S | 30.69 | 0.938 | 30.07 | 0.913 | 37.68 | 0.954 | 30.35 | 0.923 |
| + **TRAPS** | 30.72 | 0.939 | 30.14 | 0.919 | 37.74 | 0.959 | 30.40 | 0.926 |
| MIRNet | 30.54 | 0.926 | 29.76 | 0.905 | 37.17 | 0.939 | 30.21 | 0.916 |
| + **TRAPS** | 30.62 | 0.929 | 29.82 | 0.909 | 37.31 | 0.942 | 30.29 | 0.920 |
| GRL-S | 30.88 | 0.941 | 30.15 | 0.919 | 37.79 | 0.962 | 30.47 | 0.925 |
| + **TRAPS** | 30.97 | 0.947 | 30.22 | 0.922 | 37.94 | 0.968 | 30.56 | 0.931 |
| HAT-S | 30.64 | 0.937 | 29.97 | 0.911 | 37.52 | 0.944 | 30.29 | 0.917 |
| + **TRAPS** | 30.77 | 0.939 | 30.05 | 0.915 | 37.63 | 0.948 | 30.36 | 0.924 |

## 5 ABLATION STUDIES AND DISCUSSIONS

In this section, we first conduct ablation studies to investigate the impact of when to early-stop our training strategy, **TRAPS**, on the overall optimization process. Subsequently, we provide a comprehensive discussion and analysis of the properties of TRAPS from the perspectives of data scale, model size, and training duration. Then, we point out some limitations of the TRAPS.

### 5.1 THE IMPACT OF WHEN TO EARLY-STOP

We emphasize again that our training strategy models the IR task as an optimization problem from intra-pixel and inter-pixel perspectives, i.e., we first guide the model to learn the priori distribution $q(\mathcal{Y}_{p2p}|\mathcal{X}_{p2p})$ and early-stop it at an appropriate time, and then enable the model to proceed to learn the distribution $q(\mathcal{X}_{out}|\mathcal{X}_{p2p}, \mathcal{X}_{in})$. To investigate the effect of when to early-stop our training strategy on the generalization ability, we conduct the following ablation study on the color image denoising task with $\sigma = 50$. All experimental results were obtained by training with the DIV2K (Agustsson & Timofte, 2017) training set after 300 epochs and testing on four benchmark test sets (Zhang et al., 2011; Martin et al., 2001; Huang et al., 2015; Franzen, 1999), see Fig. 6.

It can be observed that in nearly all cases, as long as our training strategy is applied early in the optimization process, performance improvements are consistently observed across all test sets. The only exception is when HAT-S (Chen et al., 2023b) stops prematurely at the 4th epoch. We attribute this outlier to the fact that our strategy is regarded as an initialization method during the early training phase, where the optimization trajectory of a given model is random and unpredictable. The introduction of our strategy may cause the initial parameter updates to fall into a suboptimal local geometry landscape. However, with the continual guidance of this inductive bias, the model's opti-

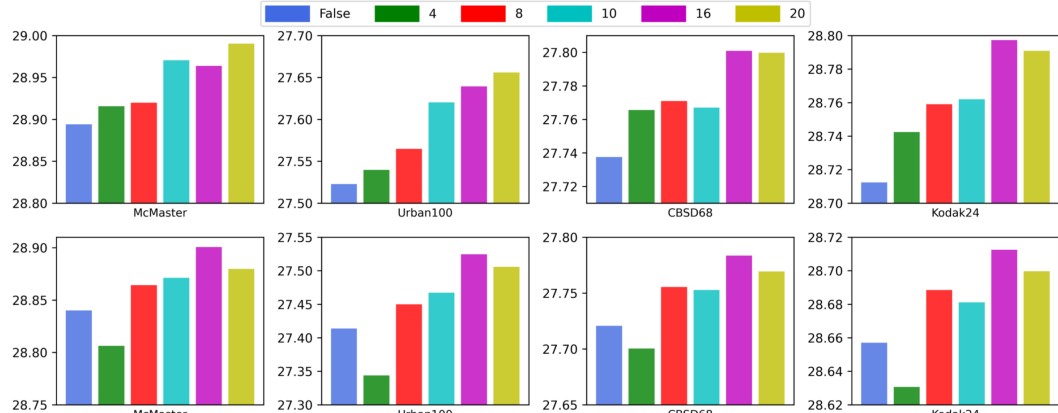

Figure 6: The color image denoising task with $\sigma = 50$. The two rows represent the experimental results of GRL-S(Li et al., 2023b) and HAT-S(Chen et al., 2023b) on the four benchmark test sets, respectively. The PSNR values are reported in RGB space. The **'False'** in the legend denotes **not** enabling our training strategy, and the number represents how many epochs to continue our training strategy before stopping it early.

mization trajectory will rapidly escape from this landscape, steering towards a better direction and yielding better generalization ability. It is also worth noting that our strategy is even able to improve GRL-S (Li et al., 2023b) and HAT-S over 0.1 dB for free on the dataset with a much more restoration difficulty, Urban100, respectively.

In addition, for HAT-S, when our training strategy early stops at the 20th epoch instead of the 16th, a slight and consistent performance decline across all test sets is observed, which is not the case for GRL-S. This result can be attributed to the fact that HAT-S has more parameters than GRL-S (9.41 M v.s. 3.12 M). More parameters indicate a stronger ability to capture the priori distribution. That is, for HAT-S, the priori distribution $q(\mathcal{Y}_{p2p}|\mathcal{X}_{p2p})$ has already been learned well enough during the first 16 epochs, at which point the model already has the ability to learn the distribution $q(\tilde{\mathcal{X}}_{out}|\mathcal{X}_{p2p}, \mathcal{X}_{in})$. Due to page limitations, more ablation studies and analyses can be found in the Appendix.

## 5.2 DISCUSSIONS AND ANALYSES

In this subsection, we comprehensively discuss and analyze the properties of the **TRAPS**.

**The Impact of Model Scales and Training Duration.** To investigate the performance improvement brought by introducing our training strategy into the existing method in relation to the model scale and training duration, we conduct the following experiments: taking SwinIR (Liang et al., 2021) and its variants as the research targets, we use the same dataset and experimental settings for the color image denoising task with $\sigma = 50$ and report the PSNR metrics on the test set Kodak24 (Franzen, 1999). The results are illustrated in Fig. 7. It can be observed that when TRAPS is introduced, the generalization ability of existing methods is improved indeed. That is, the obtained performance improvement is consistent and stable and is not suppressed by the training schedule. Specifically, Fig. 7(a) and (b) reveal the performance impact on the denoising algorithm before and after the introduction of our training strategy for different training durations. It can be seen that for SwinIR-S(Liang et al., 2021), 300 epochs of training have already made the algorithm converge under our experimental setup, and the introduction of TRAPS can achieve a steady performance improvement, and the convergence occurs earlier, i.e., it converges faster. When doubling the training duration to 600 epochs, the above observation still exists, i.e., the gains from the introduction of TRAPS do not diminish with the increase in training duration.

In addition, Fig. 7(a) and (c) unveil the sensitivity of TRAPS to different model scales. It can be observed that even when the model size is enlarged several times (2.24 M v.s. 11.5 M), our strategy still guarantees a stable and consistent PSNR improvement. Further, the performance gains are even more significant when TRAPS is introduced to a smaller model, which sheds light on the potential of deploying our strategy to real-world resource-constrained end-side scenarios. Note that in the early

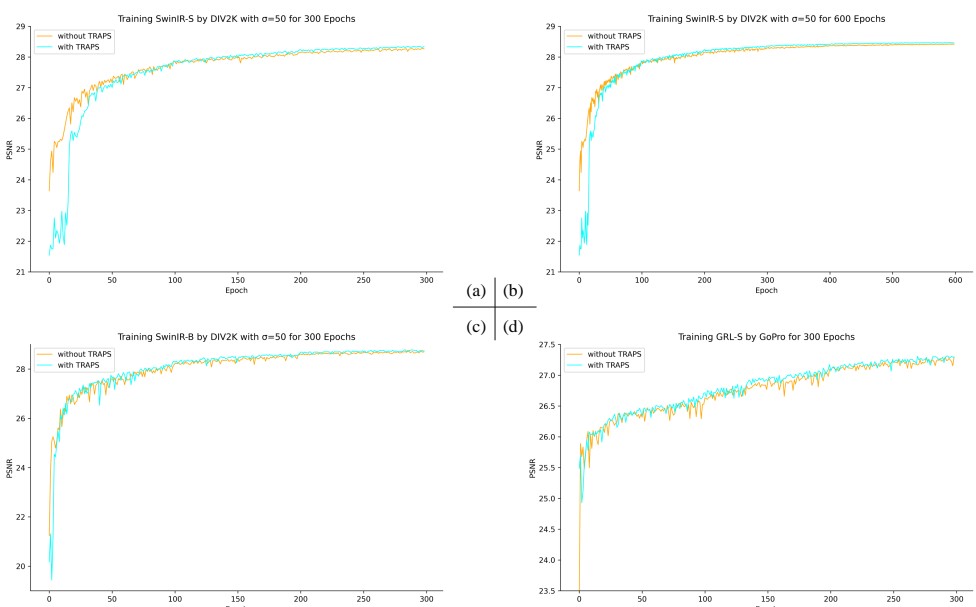

Figure 7: Performance curves for different training durations and model scales with/without the introduction of TRAPS. (a), (b) and (c) train on the DIV2K (Agustsson & Timofte, 2017) training set, and test on Kodak24 (Franzen, 1999). The PSNR values are reported in RGB space. (d) trains on the GoPro (Nah et al., 2017) training set. Since the GoPro test set is too large, we randomly sample 30 pairs of images in it as a test subset and also report the PSNR values in RGB space.

stages of training, when the epoch is small, whether TRAPS is introduced will bring about a larger gap in generalization ability for the model, i.e., the PSNR is much lower with TRAPS introduced. This is because, at this time, the model focuses on learning the priori distribution $q(\mathcal{Y}_{p2p}|\mathcal{X}_{p2p})$ rather than the PSNR-oriented distribution $q(\mathcal{X}_{out}|\mathcal{X}_{in})$. Even if the PSNR metrics are low at this time, when the model has already learned the priori distribution and shifted to learn the mapping distribution of the content and patterns within a pixel's neighborhood, the generalization ability will rapidly boost and outperform the original training approach.

Furthermore, the above observation can also be seen in the image deblurring task, as shown in Fig. 7(d). Note that the reason for such large fluctuations in the performance curves is that the test set of GoPro (Nah et al., 2017) contains too many image pairs (2103 pairs for training, and 1111 pairs for testing). In order to reduce the total training duration, just to illustrate similar phenomena, we randomly sample 1000 and 30 pairs of images in the whole train and test set as a subset, respectively. Even though the two subsets lack diversity compared to the original sets, such a setup can still reflect the actual experimental results due to the fair test configuration.

**The Impact of Data Scales.** Inspired by the observation that the performance of smaller models improves more after the introduction of TRAPS, we further investigate another factor that is crucial for training deep IR models, i.e., how the data scale will affect the performance of a given method with/without the introduction of TRAPS. To this end, we random-sample 100, 200, and 400 images from the DIV2K (Agustsson & Timofte, 2017) training set to create the DIV2K-100, DIV2K-200, and DIV2K-400 subsets, respectively, and the original 800-image set to form a total four training sets with different data scales. We train SwinIR-S (Liang et al., 2021) on the color image denoising task with the same noise level $\sigma = 50$ and the training duration set to 300 epochs. All other experimental setups remain identical. PSNR values of Kodak24 (Franzen, 1999) are reported.

The experimental results are illustrated in Fig. 8. It can be observed that, first, TRAPS provides consistent performance gains for the model under different training data scales. Furthermore, we found that the introduction of TRAPS consistently boosts the generalization ability, and this improvement remains stable regardless of changes in model size or data scale. In addition, when the data scales are decreasing successively, there will be a significant increase in the final gain from introducing TRAPS, as shown from Fig. 8(d) to (a). This observation can be attributed to the fact that our strategy itself holds the effect of data augmentation, and this effect will be more remarkable when the

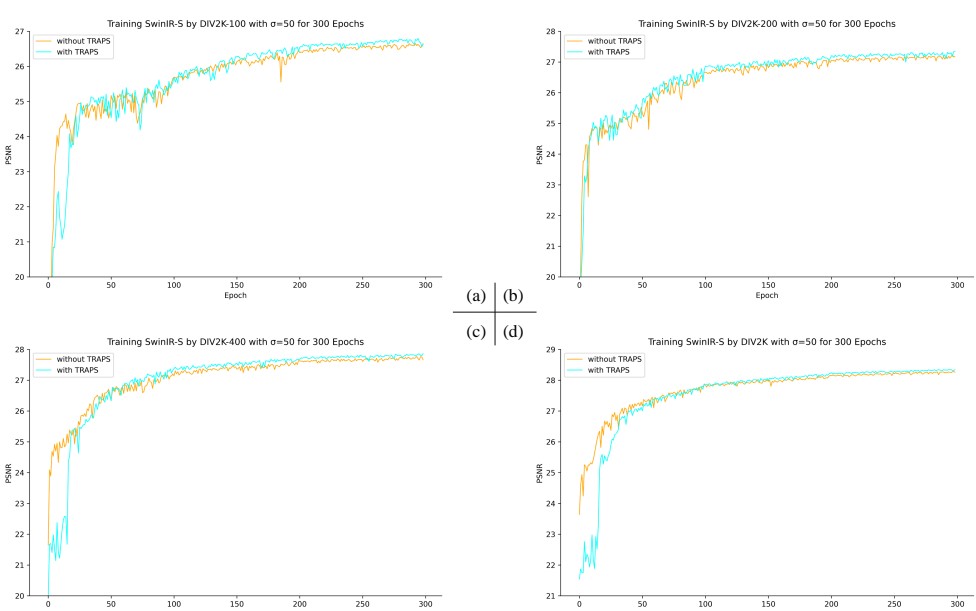

Figure 8: The experimental results on how data scale affects the performance with/without the introduction of TRAPS. We random-sample 100/200/400 image pairs from the full DIV2K (Agustsson & Timofte, 2017) training set to form the DIV2K-100/-200/-400 training subsets with different scales and uniformly train the SwinIR-S (Liang et al., 2021) on the color image denoising task with the same noise $\sigma = 50$ for 300 epochs. The PSNR values of Kodak24 (Franzen, 1999) are reported in the RGB space.

data scale is small, which further reveals the possibility that TRAPS can be deployed in some real-world scenarios where data collection is difficult. Yet, increasing the scale of the training set (e.g. DIV2K-400) can be considered equivalent to a form of data augmentation in itself, especially when compared to smaller training sets (e.g. DIV2K-100). Therefore, it is acceptable that TRAPS brings slightly lower performance gains when introduced into training scenarios with larger data scales.

**Limitation.** Although we have successfully validated the effectiveness of our strategy on several representative IR tasks, this represents only the tip of the iceberg for the broader low-level vision community, with many tasks yet to be explored. Furthermore, more properties of TRAPS have not been fully explored yet. For instance, the optimal moment to activate TRAPS: if the training process before and after each learning rate change is treated as distinct sub-problems within the whole optimization problem, it would be interesting to investigate the outcomes of activating TRAPS separately at the early stages following each learning rate adjustment.

## 6 CONCLUSION

In this paper, we propose a concise and elegant training strategy for the IR task, which adheres to the idea of 'Haste makes waste' and abandons the paradigm that existing IR methods directly learn the pattern distribution between degraded/clean images. The new paradigm decomposes the IR optimization process into a distribution mapping problem from two perspectives. We propose TRAPS to guide the model to first learn an a priori distribution for intra-pixel mapping. Conditioned on this prior, the model is allowed to learn the mapping distribution among inter-pixel patterns. The newly learned distribution can be regarded as a joint distribution, which smoothly transfers the final distribution from the pixel level to the pattern one. Compared to the original method, the learned joint distribution is closer to the ideal one, yielding a stronger representation capability.

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

# A   Appendix

We provide more supporting materials in this supplementary file. First, we pronounce the overall training protocol. Then, for further convenience of the reader, we give the pseudocode of the algorithm for TRAPS. Finally, more experimental results are presented.

## A.1   Overall training settings

In all experiments, we use the following training settings, unless mentioned otherwise. We re-train and test all the models on 8 Nvidia Tesla V100 GPUs using DataParallel encapsulation mode. The specific settings are as follows: The training is with Adam optimizer ($\beta_1 = 0.9$, $\beta_2 = 0.999$) and L1 loss for all models with the initial learning rate $1 \times 10^{-4}$ gradually halved for every 100 epochs and we clip the norm of the gradient with max norm 10. For fair comparisons, the patch size and batch size are set to 64 and 16, respectively. For training scenarios with smaller data scales, we use a smaller learning rate to ensure the models' convergence.

## A.2   The pseudocode of TRAPS

The pseudocode for TRAPS is as follows:

---

**Algorithm 1** In**TRA**-Patch **P**ixel **S**huffle (TRAPS)

---

**Input:** Two semantic preserved image batches, images1/images2, with the shape of (B, C, H, W).
    # B: the batch size, C: the number of channels, H/W: the dimensions of the image.
  1: Flatten each image along the space dimension into 2D tensors of size (C, (HW)).
  2: Generate random shuffle indices for each image in the batch, with the size (B, C, (HW)).
  3: Use the random indices to shuffle the flattened image data.
  4: Reshape the shuffled images back into their original shape (B, C, H, W).
**Output:** Two semantic stripped image batches, shuffled_images1/shuffled_images2, with the shape of (B, C, H, W).

---

Notably, since the shuffle indices are generated randomly each time, this ensures the randomness of pixel spatial position changes in the resulting semantically stripped images. Consequently, the priori distribution $q(\mathcal{Y}_{p2p}|\mathcal{X}_{p2p})$ learns from more independent mappings of individual pixels.

## A.3   Additional experimental results

Table 3: The impact of early stopping TRAPS on SwinIR-S (Liang et al., 2021).The results are about the color image denoising task with $\sigma = 50$. The PSNR values are reported in RGB space. The best results are **bolded**. 'False' denotes without the introduction of TRAPS.

| When Stops | McMaster | Urban100 | CBSD68 | Kodak24 |
|:---:|:---:|:---:|:---:|:---:|
| False | 28.17 | 28.88 | 27.45 | 28.30 |
| 4 | 28.17 | 26.89 | 27.44 | 28.30 |
| 8 | 28.15 | 26.90 | 27.45 | 28.30 |
| 10 | 28.19 | 26.92 | 27.45 | 28.32 |
| 16 | **28.25** | **26.95** | **27.49** | **28.36** |
| 20 | 28.23 | 26.90 | 27.49 | 28.36 |
| 24 | 28.24 | 26.90 | 27.48 | 28.34 |
| 30 | 28.22 | 26.84 | 27.46 | 28.32 |
| 33 | 28.14 | 26.79 | 27.46 | 28.31 |

In this section, more quantitative results and quantitative visual comparisons are provided.

**More Ablations on the Impact of When to Early-Stop.** Table 3, Table 4, Table 5 and Table 6 provide more results on the impact of when to early-stop. The following results can be observed: 1) activating TRAPS early in the training can yield stable performance gains for the original model; 2) each model has an optimal early stopping point, while continually introducing TRAPS after this

Table 4: The impact of early stopping TRAPS on SwinIR-B (Liang et al., 2021).The results are about the color image denoising task with $\sigma = 50$. The PSNR values are reported in RGB space. The best results are **bolded**. 'False' denotes without the introduction of TRAPS.

| When Stops | McMaster | Urban100 | CBSD68 | Kodak24 |
|---|---|---|---|---|
| False | 28.88 | 27.57 | 27.79 | 28.77 |
| 4 | 28.89 | 27.60 | 27.81 | 28.79 |
| 8 | **28.94** | **27.65** | **27.83** | **28.81** |
| 10 | 28.92 | 27.63 | 27.81 | 28.79 |
| 16 | 28.84 | 27.56 | 27.77 | 28.74 |
| 20 | 28.82 | 27.54 | 27.77 | 28.73 |
| 24 | 28.83 | 27.53 | 27.77 | 28.73 |
| 30 | 28.82 | 27.50 | 27.75 | 28.72 |

Table 5: The impact of early stopping TRAPS on HAT-S (Chen et al., 2023b).The results are about the color image denoising task with $\sigma = 50$. The PSNR values are reported in RGB space. The best results are **bolded**. 'False' denotes without the introduction of TRAPS.

| When Stops | McMaster | Urban100 | CBSD68 | Kodak24 |
|---|---|---|---|---|
| False | 28.84 | 27.41 | 27.72 | 28.66 |
| 4 | 28.81 | 27.34 | 27.70 | 28.63 |
| 8 | 28.86 | 27.45 | 27.76 | 28.69 |
| 10 | 28.87 | 27.47 | 27.75 | 28.68 |
| 16 | **28.90** | **27.52** | **27.78** | **28.71** |
| 20 | 28.88 | 27.51 | 27.77 | 28.70 |
| 24 | 28.88 | 27.51 | 27.77 | 28.70 |
| 30 | 28.89 | 27.52 | 27.77 | 28.71 |

Table 6: The impact of early stopping TRAPS on GRL-S (Li et al., 2023b).The results are about the color image denoising task with $\sigma = 50$. The PSNR values are reported in RGB space. The best results are **bolded**. 'False' denotes without the introduction of TRAPS.

| When Stops | McMaster | Urban100 | CBSD68 | Kodak24 |
|---|---|---|---|---|
| False | 28.89 | 27.52 | 27.74 | 28.71 |
| 4 | 28.92 | 27.54 | 27.77 | 28.74 |
| 8 | 28.92 | 27.56 | 27.77 | 28.76 |
| 10 | 28.97 | 27.62 | 27.77 | 28.76 |
| 16 | 28.96 | 27.64 | 27.80 | 28.80 |
| 20 | **28.99** | **27.66** | **27.80** | 28.79 |
| 24 | 28.98 | 27.64 | 27.81 | **28.81** |
| 30 | 28.98 | 27.65 | 27.80 | 28.79 |

point will compromise the model's generalization ability. Therefore, at this moment, the model has learned the priori distribution $q(\mathcal{Y}_{p2p}|\mathcal{X}_{p2p})$ and has the ability to start learning the cross-pixel content and pattern mapping distribution $q(\mathcal{X}_{out}|\mathcal{X}_{p2p}, \mathcal{X}_{in})$. It is worth noting that this early stopping point is not always the same for different models due to their distinct optimization trajectories, and thus this handcrafted transition can be improved into a more adaptive approach in future work.

**Visualization of the Training Curve.** To validate the convergence after the introduction of TRAPS, we present some training curves as shown in Fig. 9, Fig. 10, Fig. 11 and Fig. 12. It can be consistently observed that, on the one hand, the training loss is higher in the early stages of training after the introduction of TRAPS than without it. This is acceptable because the architecture of the model is static and the computation attributes do not change, e.g., the sliding-window attention mechanism that focuses on pixel neighborhoods. Instead, we modify the original input, forcibly shuffling the pixels thereby guiding the model to learn semantic stripped independent pixel-to-pixel mappings. As a consequence, the computation mechanism of cross-pixel interaction will not work very well at this moment, which in turn leads to higher loss. On the other hand, after deactivating TRAPS, the

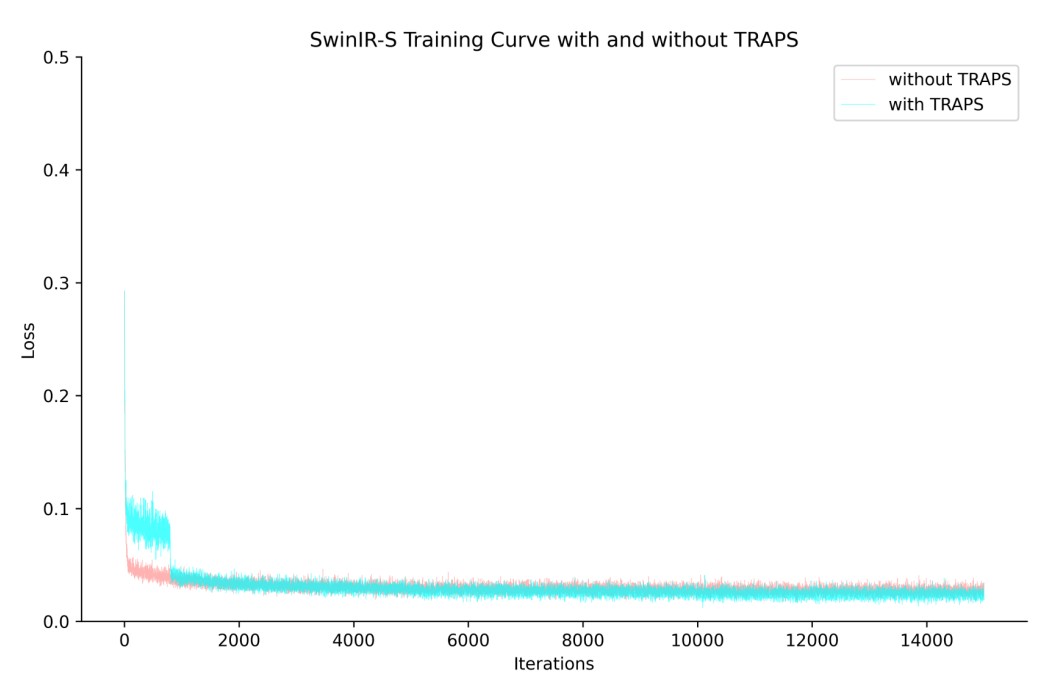

Figure 9: Training curves on SwinIR-S (Liang et al., 2021) with/without the introduction of TRAPS.

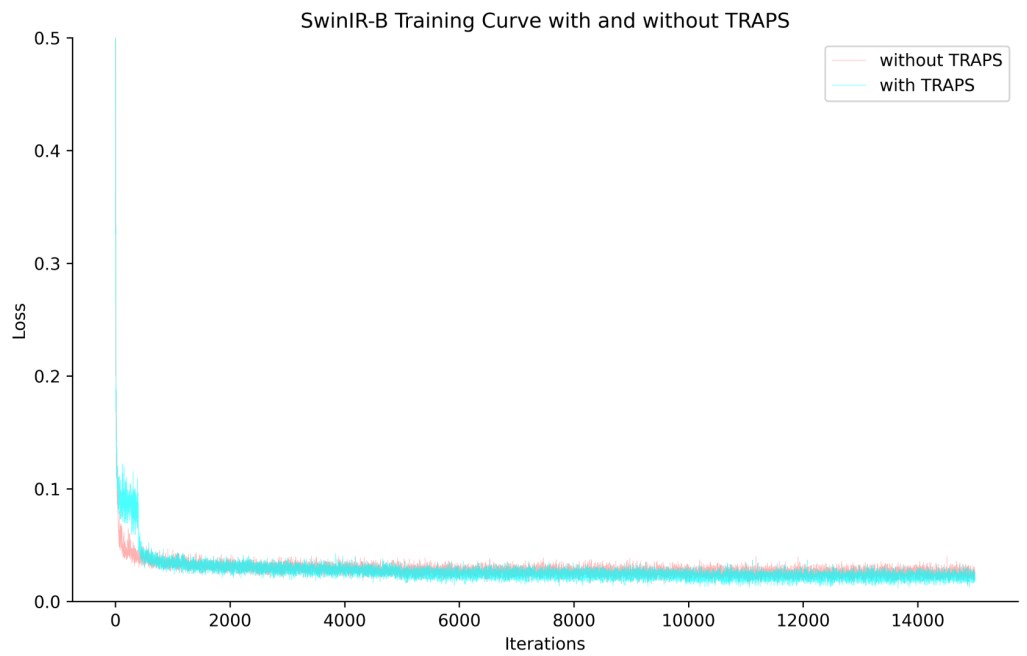

Figure 10: Training curves on SwinIR-B (Liang et al., 2021) with/without the introduction of TRAPS.

training loss is lower than the original one. This indicates that with the introduction of TRAPS, the optimization trajectory of the model is indeed steered into a better loss geometry landscape, which consequently yields a better generalization ability.

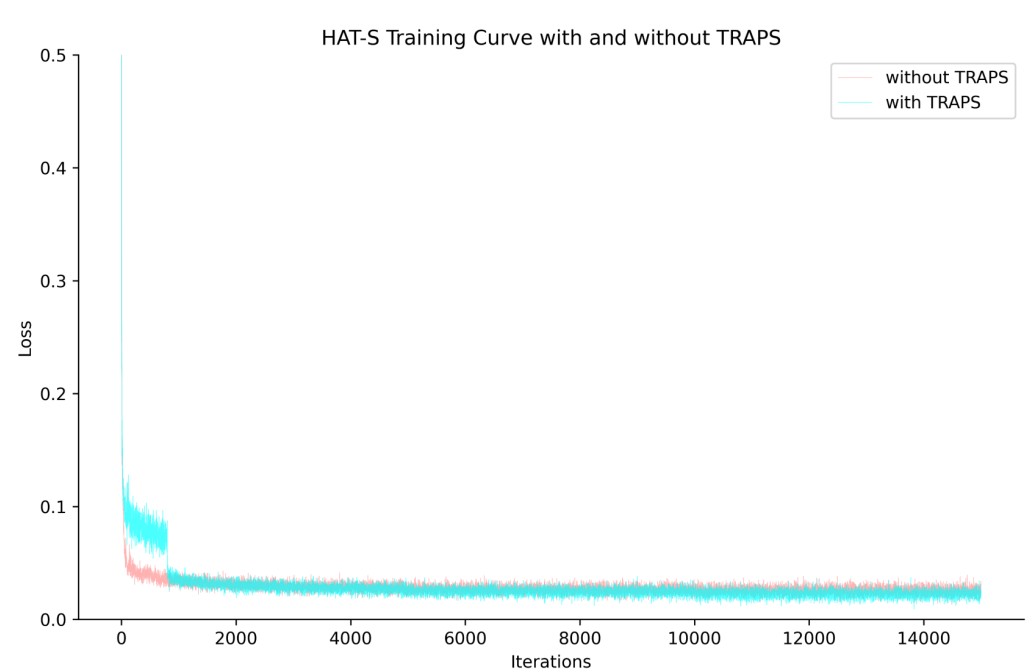

Figure 11: Training curves on HAT-S (Chen et al., 2023b) with/without the introduction of TRAPS.

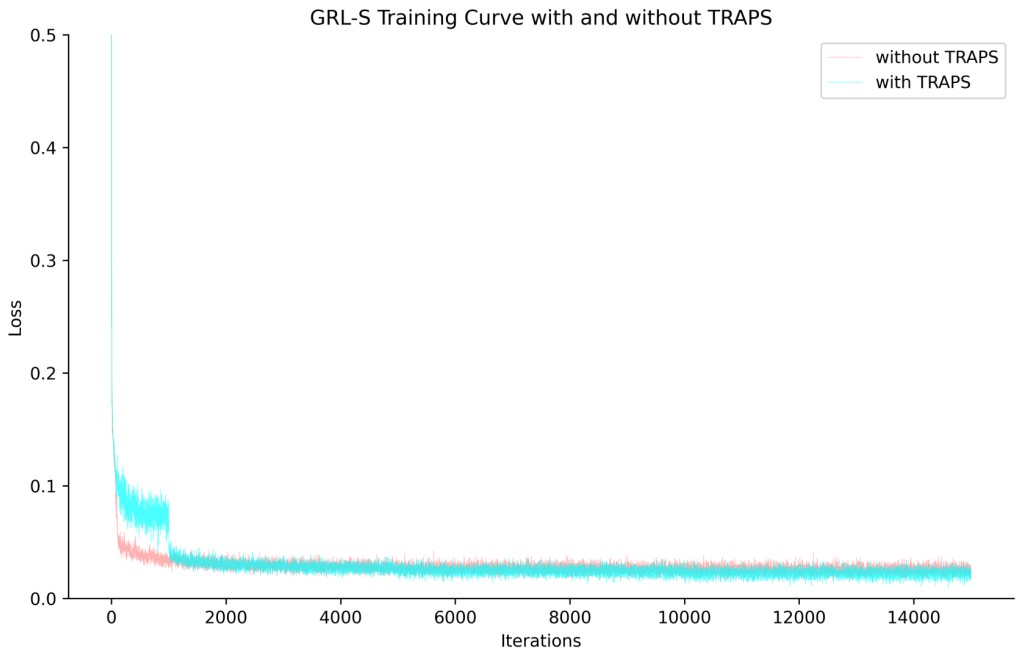

Figure 12: Training curves on GRL-S (Li et al., 2023b) with/without the introduction of TRAPS.

**Additional Visual Comparison Results under the Real Scenario.** The main manuscript only shows the experimental results on the synthetic image denoising task, to further validate the generalization ability of our approach, we directly transfer the models trained with/without TRAPS to real-world denoising scenarios. The visual comparison results are shown in Fig. 13 and Fig. 14. It can be seen that our strategy enables the model to remove more noise from the real scene and can produce more perceptually pleasing outputs.

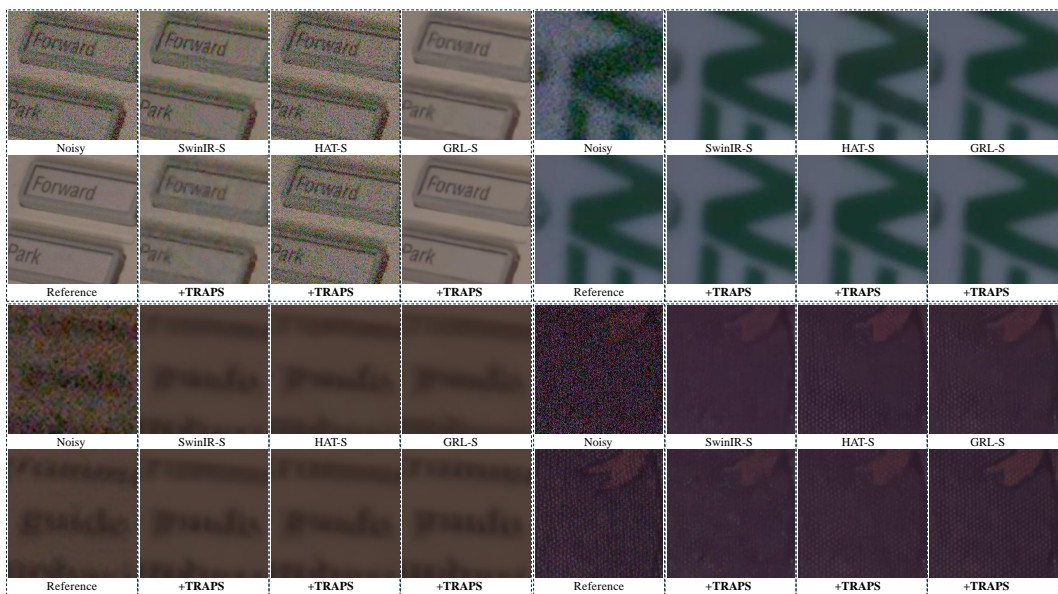

Figure 13: Visual comparisons of Real-World image noise removal on SIDD (Abdelhamed et al., 2018) dataset. Zoom in for more details.

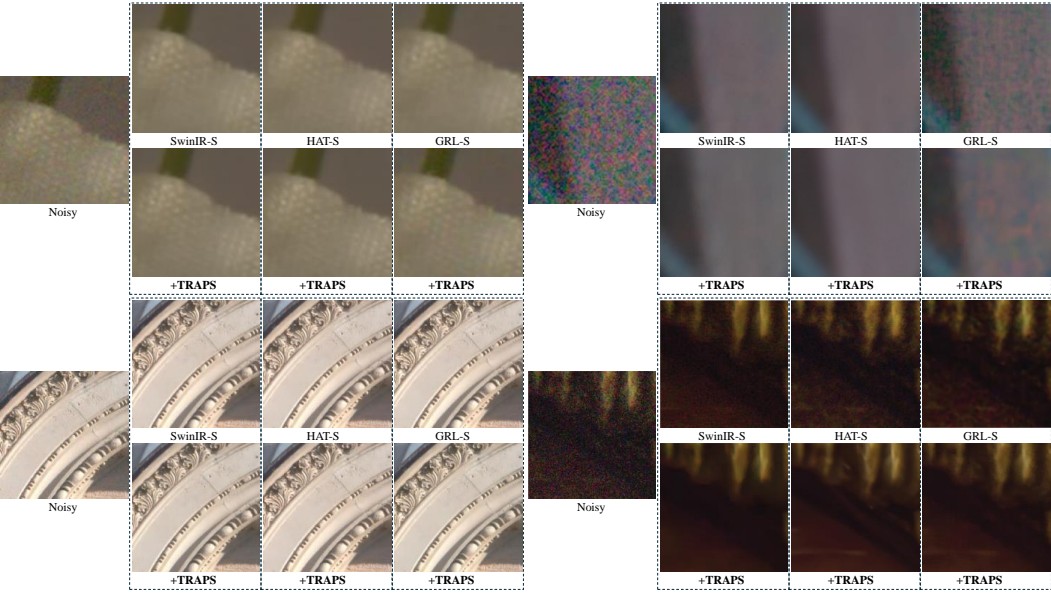

Figure 14: Visual comparisons of Real-World image noise removal on DND (Plotz & Roth, 2017) dataset. Zoom in for more details.

