# OpenReview forum: "Haste Makes Waste: Teaching Image Restoration to Learn Distributions from Pixels to Patterns"
_ICLR.cc/2025/Conference — ICLR 2025 Conference Withdrawn Submission_

### Official Review · Reviewer_NRqa · 2024-10-28

**Soundness:** 2
**Presentation:** 3
**Contribution:** 1
**Rating:** 3
**Confidence:** 4

**Summary:**

This paper proposes a new training strategy for image restoration tasks by modeling from both intra-pixel and inter-pixel perspectives. This approach enhances the network's performance without requiring additional training data or time.

**Strengths:**

The proposed approach enhances the network's performance without requiring additional training data or time.

**Weaknesses:**

1. The method lacks sufficient innovation and resembles more of a trick, which is not enough to support a paper at ICLR.
2. The authors retrained all comparison methods during their experiments, yet the reported results for these methods are significantly lower than those in the original papers. This raises uncertainty as to whether the proposed method is only effective under the specific training conditions used by the authors. Even under these conditions, the performance improvement is minimal.
3. The authors claim their method is designed for image restoration tasks, but the experiments only include image denoising and deblurring. There is a lack of experiments on other image restoration tasks, such as image super-resolution and deraining.

**Questions:**

Train the network using the same settings as the comparison method to see if the performance can surpass the comparison method.

---

### Official Review · Reviewer_CGJN · 2024-10-30

**Soundness:** 3
**Presentation:** 3
**Contribution:** 3
**Rating:** 6
**Confidence:** 3

**Summary:**

This paper revisits the infrared task by identifying its fundamental pixel-to-pixel regression nature and modeling it as an optimization problem from both intra-pixel and inter-pixel perspectives. It proposes a novel training strategy tailored to these observations, serving as a free data augmentation method or a warm-up approach for training. This paper’s strategy can be seamlessly integrated into existing supervised IR methods without additional burden, effectively introducing an inductive bias that enhances model performance.

**Strengths:**

**Enhanced Performance**: The proposed training strategy introduces an inductive bias that can significantly boost the performance of existing supervised infrared (IR) methods, leading to improved accuracy in pixel-to-pixel regression tasks.

**Seamless Integration**: This paper's strategy can be easily incorporated into current IR frameworks without requiring major modifications, allowing researchers to adopt the method with minimal effort and disruption.

**Versatile Application**: By functioning as both a free data augmentation technique and a training warm-up approach, the proposed strategy provides flexible options for enhancing model training, making it applicable in various IR contexts.

**Weaknesses:**

**Limited Early-Stage Effectiveness**: The TRAPS strategy relies on a gradual transition from intra-pixel to inter-pixel optimization, which may delay capturing complex content distributions early in training, potentially leading to slower initial convergence.

**Dependency on Pre-Generated Indices**: The need to shuffle pixels according to pre-defined indices might introduce constraints, as it requires careful setup and may limit flexibility in adapting to varying IR tasks or datasets.

**Potential for Overhead in Warm-Up Phase**: Although designed to streamline optimization, the warm-up phase might add computational overhead, as the network initially focuses on simplified pixel mappings, which could lengthen the overall training duration in some cases.

**Questions:**

See Weaknesses

---

### Official Review · Reviewer_EVEX · 2024-11-03

**Soundness:** 3
**Presentation:** 2
**Contribution:** 3
**Rating:** 5
**Confidence:** 5

**Summary:**

This paper introduces a new training strategy for image restoration (IR) tasks. The strategy, named TRAPS (InTRA-patch Pixel-Shuffle), addresses the IR problem by modeling it as a distribution mapping challenge from two perspectives: intra-pixel regression and inter-pixel interaction. The method starts by teaching the model to learn a simpler pixel-by-pixel distribution, which serves as a prior and inductive bias, and then transitions to learning cross-pixel pattern distributions. The proposed approach aims to improve the model's ability to learn complex pattern mappings between degraded and clean images by breaking down the learning process into more manageable stages.

**Strengths:**

1. The paper proposes a novel training approach that addresses the complexity of learning pattern distributions in IR by breaking it down into simpler stages, which is a creative solution to a known challenge in the field.

2. The method is evaluated extensively on benchmark datasets, demonstrating consistent improvements across different models and tasks, which speaks to the robustness of the approach.

3. TRAPS can be integrated into existing supervised IR methods without additional burden, making it a versatile tool that can potentially benefit a wide range of IR models.

4. The paper provides a theoretical justification for the training strategy by modeling the IR task as an optimization problem involving distribution mapping, which adds depth to the understanding of IR processes.

**Weaknesses:**

1. The paper does not discuss the potential for overfitting, especially since the model is learning from a shuffled pixel distribution, which could lead to different characteristics compared to natural image statistics.

2. Although the method shows good prospects in IR tasks, it is not clear how well it can be generalized to other low-level vision tasks. Because the method proposed by the author is very simple to implement and the theory is simple, sufficient experiments are the premise to prove its effectiveness. At present, only two tasks do not seem to be enough to prove its effectiveness and scalability. Other common restoration tasks are also necessary, including: image super-resolution, image dehazing, image deraining, low-light enhancement, etc.

3. The method proposed in the article is simple and effective. But what is its computational cost for the network? If the repair network is larger, will the computational cost of this method also increase? This part should be further analyzed.

4. Are there other visualizations and more detailed theoretical justifications that could further support the proposed optimization directions?

Since the article is quite interesting and has a new perspective, I will make corresponding changes based on the author's rebuttal.

**Questions:**

None

---

### Note · Authors · 2024-11-14

**Comment:**

We sincerely appreciate the detailed and constructive feedback from all three reviewers. However, certain technical details of our work may not have been fully appreciated. After careful consideration, we have decided to withdraw our paper.

**Withdrawal Confirmation:**

I have read and agree with the venue's withdrawal policy on behalf of myself and my co-authors.